# Garment4D: Garment Reconstruction
# from Point Cloud Sequences

**Fangzhou Hong[1], Liang Pan[1], Zhongang Cai[1,2,3], Ziwei Liu[1]✉**
[1]S-Lab, Nanyang Technological University    [2]SenseTime Research    [3]Shanghai AI Laboratory
{fangzhou001, liang.pan, ziwei.liu}@ntu.edu.sg   caizhongang@sensetime.com

## Abstract

Learning to reconstruct 3D garments is important for dressing 3D human bodies of different shapes in different poses. Previous works typically rely on 2D images as input, which however suffer from the scale and pose ambiguities. To circumvent the problems caused by 2D images, we propose a principled framework, **Garment4D**, that uses 3D point cloud sequences of dressed humans for garment reconstruction. Garment4D has three dedicated steps: sequential garments registration, canonical garment estimation, and posed garment reconstruction. The main challenges are two-fold: **1)** effective 3D feature learning for fine details, and **2)** capture of garment dynamics caused by the interaction between garments and the human body, especially for loose garments like skirts. To unravel these problems, we introduce a novel Proposal-Guided Hierarchical Feature Network and Iterative Graph Convolution Network, which integrate both high-level semantic features and low-level geometric features for fine details reconstruction. Furthermore, we propose a Temporal Transformer for smooth garment motions capture. Unlike non-parametric methods, the reconstructed garment meshes by our method are separable from the human body and have strong interpretability, which is desirable for downstream tasks. As the first attempt at this task, high-quality reconstruction results are qualitatively and quantitatively illustrated through extensive experiments. Codes are available at https://github.com/hongfz16/Garment4D.

## 1   Introduction

Garment reconstruction is a crucial technique in many applications, *e.g.* virtual try-on [1], VR/ AR [2] and visual effects [3]. Extensive efforts [4–15] have been put into reconstructing the human body and garments as a whole with the help of implicit representation or volumetric representation. However, it is desirable to have a controllable garment model in many applications. In this work, we focus on the parametric modeling of 3D garment, which has advantages in the following two ways. Firstly, we can separate garments from human body. Secondly, the topology of the reconstructed meshes can be controlled, and therefore allows downstream tasks that require high interpretability.

Different from previous parametric methods for garment reconstruction [7, 8, 16, 17] that take 2D RGB images as input, we choose to approach the problem from the perspective of 3D, specifically point cloud sequences (shown in Fig. 1), for the following three reasons. Firstly, 3D inputs eliminate scale and pose ambiguities that are difficult to avoid when using 2D images. Secondly, exploiting temporal information is important for garment dynamics capturing, at which there are few attempts. Thirdly, recent development in 3D sensors (*e.g.* LiDAR) has reduced the cost and difficulties in obtaining point clouds, which makes it easier to leverage 3D point clouds for research problems and commercial applications. However, garment reconstruction from point cloud sequences has not been properly explored by previous works. We contribute an early attempt at this meaningful task in this work. Our advantages against other garment reconstruction methods are listed in Tab. 1.

35th Conference on Neural Information Processing Systems (NeurIPS 2021).

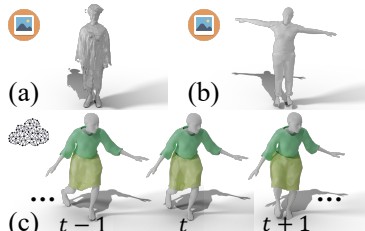

| Method Type | (a) Template-Free | (b) SMPL+D | (c) Garment4D |
|---|---|---|---|
| Input Modality | image | image | pcd seq. |
| No Ambiguity | ✗ | ✗ | ✓ |
| Separable | ✗ | ✗* | ✓ |
| Interpretability | ✗ | ✓ | ✓ |
| Loose Garments | ✓ | ✗ | ✓ |

**Figure 1 & Table 1: Comparison With Popular Garment Reconstruction Methods.** Two major types of garment reconstruction methods, *i.e.* **(a)** template-free and **(b)** SMPL+D-based methods, are compared to **(c)** Garment4D in terms of input modality, scale/ pose ambiguity, ability to separate garments from body, mesh interpretability and ability to deal with loose garments. *Some modified version of SMPL+D [16] could separate garments from body. 'pcd seq.' stands for point cloud sequence.

The proposed garment reconstruction framework, Garment4D, consists of three major parts: **a)** sequential garments registration, **b)** canonical garment estimation and **c)** posed garment reconstruction. In the registration part, we have several sequences of garment meshes which have different mesh topology across the sequences but share same topology inside one sequence. For each type of garments (*e.g.* T-shirt, trousers, skirt), we use an optimization-based method to register one frame from each sequence to a template mesh. Then, inside each sequence, a barycentric interpolation method is used to re-mesh other frames to unify their topology with the template mesh. Thereafter, as the first step of garment reconstruction, following the practice of previous parametric methods [16, 17], we estimate the canonical garment mesh for each sequence by a semantic-aware garment PCA coefficients encoder, which takes the dressed human point cloud sequences as input.

The posed garment reconstruction part is where the challenges come in for two reasons. Firstly, due to unordered and unstructured nature of point clouds, it is non-trivial to learn low-level geometric features directly. Secondly, it is challenging to capture garment dynamics caused by interactions between the human body and garments. Especially, it is difficult to model the non-rigid deformation of loose garments (*e.g.* skirts), which depends on both current human pose and previous human motions. To address the challenges, we first apply the Interpolated Linear Blend Skinning (LBS) to the estimated canonical garments as proposals. Unlike [7, 8, 16] that depends on SMPL+D model to skin the garment, the proposed Interpolated LBS has the ability to perform skinning on loose garments without artifacts. Moreover, our method does not require learning, which is different from [17]. After obtaining the proposals, we present the Proposal-Guided Hierarchical Feature Network along with the Iterative Graph Convolution Network (GCN) for efficient geometric feature learning. Meanwhile, a Temporal Transformer is utilized for temporal fusion to capture smooth garment dynamics.

Our contributions can be summarised as:

**1)** We propose a practical pipeline for the reconstruction of various garments (including challenging loose garments) that are driven by large human movements, which does not rely on any particular human body parametric model or extra annotations other than human and garment meshes.

**2)** Many novel learning modules, such as Proposal-Guided Hierarchical Feature Network, Iterative GCN, and Temporal Transformer are proposed, which effectively learn spatio-temporal features from 3D point cloud sequences for garment reconstruction.

**3)** We establish a comprehensive dataset consisting of $2,300$ 3D point cloud sequences by adapting CLOTH3D [18], which serves as the first benchmark dataset for evaluating posed garment reconstruction based on point cloud sequences.

In addition, this work presents an early attempt at exploiting temporal information for capturing garment motions. To the best of our knowledge, we are the first to explore garment reconstruction from point cloud sequences.

## 2 Our Approach

We introduce our approach, Garment4D, in this section. Firstly, our garment model is formulated in Subsection 2.1. Afterwards, we elaborate the three major steps of Garment4D: 2.2 sequential

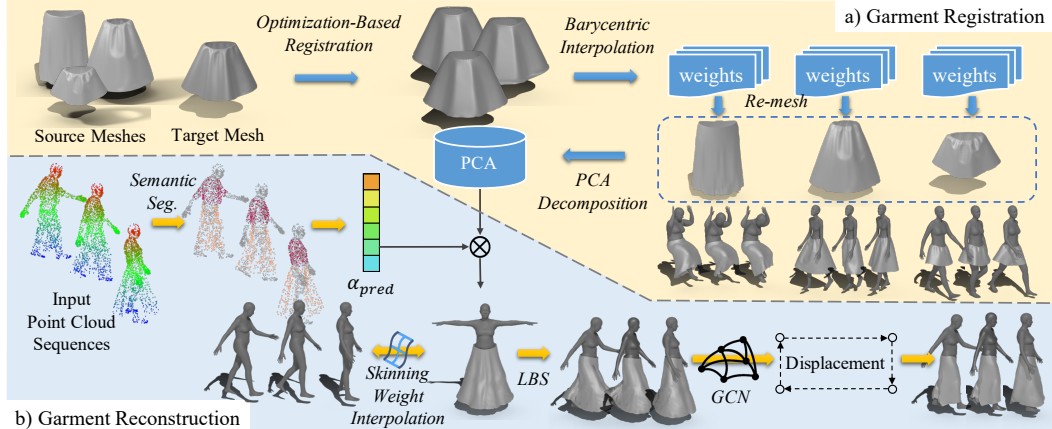

**Figure 2: Garment4D Pipeline Overview. a)** The upper part illustrates the garment registration. Source meshes are registered to target meshes and re-meshed using barycentric interpolation. **b)** The lower part shows the garment reconstruction pipeline. The first stage predict PCA coefficients to estimate the canonical garment. The second stage predicts displacements guided by the interpolated LBS proposals.

garments registration, 2.3 canonical garment estimation, and 2.4 posed garment reconstruction. Training loss functions are introduced in Subsection 2.5. An overview of the proposed Garment4D is shown in Fig. 2.

## 2.1  3D Garment Model

We design the garment model by the following three parts:

**Canonical Garment Model.** Formally, the canonical garment $T \in \mathbb{R}^{N \times 3}$ can be formulated as $T(\alpha) = G + C\alpha$ , where $G$ represents the mean shape of current type of garment, $C$ is the PCA components that form garment shape sub-spaces, $\alpha$ represents the PCA coefficients of the corresponding garment.

**Interpolated LBS.** One critical problem in skinning step is to get a set of skinning weights of garment vertices that can provide reasonable proposals of posed garments. Previous methods [7, 8, 16] approximate the garment skinning weights by using the weights of the closest human mesh vertex. However, these methods can result in mesh artifacts, and hence limit the quality of reconstructed garments for the following two facts. Firstly, the resolutions of our garment meshes is higher than that of human meshes. Secondly, loose garments like skirts do not always share same topology with human body. To avoid artifacts, we propose to interpolate the skinning weights from $K$ nearest neighbors of human mesh vertices $T_h$. Laplacian smoothing is also performed to further ensure the smoothness of the skinning weights.

**Displacement Prediction.** For those challenging loose garments like skirts and dresses, the interpolated results are mostly not satisfactory. To this end, we further refine the results of interpolated LBS by predicting displacements $D$. In general, the whole garment model can be formulated as

$$M(\alpha, \theta, K, \mathcal{W}_h, D) = W\big(T(\alpha), J, \theta, \mathcal{W}_g(\alpha, T_h, \mathcal{W}_h, K)\big) + D, \tag{1}$$

$$\mathcal{W}_g(\alpha, T_h, \mathcal{W}_h, K) = \mathcal{W}_g^K\big(T(\alpha), T_h, \mathcal{W}_h, K\big) I_w\big(T(\alpha), T_h, K\big), \tag{2}$$

where $\theta \in \mathbb{R}^{|J| \times 3}$ represents the axis-angle of each joint, $\mathcal{W}_h \in \mathbb{R}^{M \times |J|}$, $\mathcal{W}_g \in \mathbb{R}^{N \times |J|}$ are the skinning weights of human and garment mesh vertices, $J$ is the joint locations of human body, $\mathcal{W}_g^K \in \mathbb{R}^{M \times |J| \times K}$ is the garment skinning weights of $K$ nearest body vertices, $I_w \in \mathbb{R}^{M \times K}$ is the inverse distance weights used for interpolation.

## 2.2  Garment Registration

In order to perform PCA decomposition to the same type of garments, we unify the mesh topology by registering the raw meshes to a template mesh $T$. A visualization of the registration process is illustrated in Fig. 2 (a).

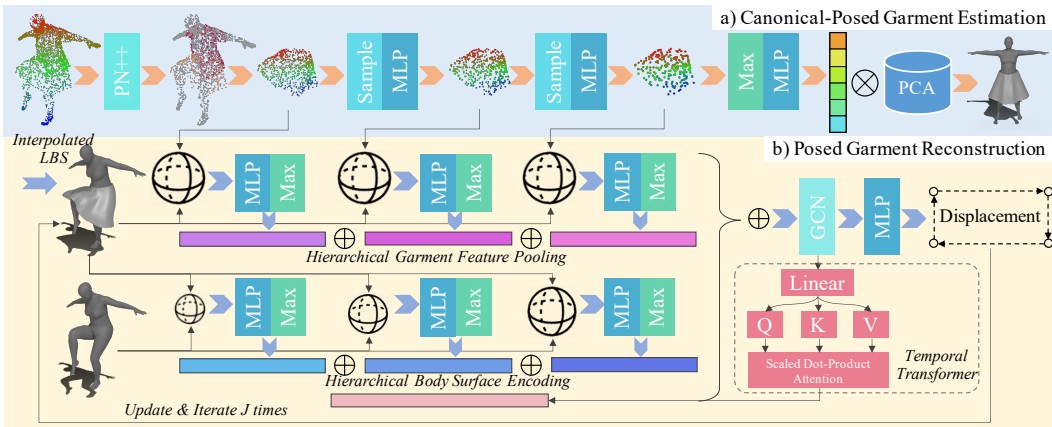

**Figure 3: Details of Garment Reconstruction Network. a)** The canonical garment estimation takes point clouds as input. Semantic segmentation is performed first to get the garment point clouds, upon which down-sampling and feature extraction are further applied to predict the PCA coefficients. **b)** Using the interpolated LBS, proposals of posed garments are obtained, guided by which, a Hierarchical Feature Network along with Iterative GCN and Temporal Transformer are used to iteratively predict per-vertex displacements.

Considering that source meshes have large shape variance and diverse sleeve/ hem length, we use a boundary-aware optimization method to align the source meshes to the target mesh. The registration loss function $L_{reg}$ can be formulated as

$$\mathcal{L}_{reg} = \lambda_c \mathcal{L}_{CD}(R', T) + \lambda_e \mathcal{L}_{EL}(R') + \lambda_n \mathcal{L}_{NC}(R') + \lambda_b \mathcal{L}_{CD}^B(R', T), \qquad (3)$$

where $\mathcal{L}_{CD}(R', T)$ is the Chamfer Distance (CD) between source and target mesh vertices, $\mathcal{L}_{EL}(R')$ normalizes Edge Length of the source mesh, $\mathcal{L}_{NC}(R')$ maintains the Normal Consistency between neighbouring faces, $\mathcal{L}_{CD}^B(R', T)$ minimizes the CD between corresponding Boundaries of source and target meshes (*e.g.* cuffs and neckline), and $\lambda_*$ are corresponding weights for each loss term. The $\mathcal{L}_{CD}^B$ term makes the optimization process aware of the boundaries of the meshes, which is essential for aligning garments with large shape variance.

For each sequence, the aligned garment mesh $R' \in \mathbb{R}^{X \times 3}$ is then used to calculate barycentric interpolation weights against the vertices of template mesh $T$, which is further used to re-mesh garment meshes in other frames. Specifically, for each vertex $x$ of $T$, the nearest face $f$ in $R'$ is found. We project $x$ to $f$ and then calculate the barycentric coordinate $(w_1, w_2, w_3)$ of the projected point $x'$ on $f$. Using the barycentric coordinates, it is straight-forward to re-mesh all the other frames of garment meshes to the topology of the template mesh $T$.

## 2.3 Canonical Garment Estimation

To estimate canonical garments $T(\alpha)$, we predict the PCA coefficients $\alpha$ by using the segmented 3D points of garments from input point cloud sequences $P \in \mathbb{R}^{\mathcal{T} \times p \times 3}$, where $\mathcal{T}$ is the sequence length. As shown in Fig. 3 (a), we first perform point-wise semantic segmentation to select the 3D points of garments for each frame, which results in a sequence of garment point clouds $P_g \in \mathbb{R}^{\mathcal{T} \times p_g \times 3}$. Subsequently, $P_g$ is used to predict the PCA coefficients $\alpha$ for each sequence, which is further used to reconstruct target canonical garments by $T(\alpha) = G + C\alpha$.

## 2.4 Posed Garment Reconstruction

**Proposal-Guided Hierarchical Feature Network.** As introduced in Subsection 2.1, the skinned garments $M_p = W(T(\alpha), J, \theta, \mathcal{W}_g)$ of the proposed garment model can provide reasonable initial proposals for posed garments. However, due to complex human motions and garment flexibility, especially for intense movements and loose skirts, the initial proposals of garment poses can be far different from real poses. To address these problems, we propose a novel Proposal-Guided Hierarchical Feature Network, consisting of two major parts: 1) Hierarchical Garment Feature Pooling, and 2) Hierarchical Body Surface Encoder. Displacement $D$ is predicted to refine the initial garment poses. Detailed architectures of these two modules are demonstrated in Fig. 3 (b).

The proposed Hierarchical Garment Feature Pooling can effectively capture local geometric details from garments. Hierarchical Body Surface Encoder facilitates predicting correct garment poses by considering interactions between garment and human body. By estimating proper contacts between garments and skin, our body surface encoder highly alleviates the interpenetration problems.

- **Hierarchical Garment Feature Pooling** aims at pooling accurate geometric features and rich semantic information for each vertex of mesh proposals. Specifically, for each vertex of $M_p$, several query balls with different radii are constructed to sample points from the segmented point clouds of garments. To make use of the rich semantic information extracted when predicting the PCA coefficients $\alpha$, different levels of down-sampled garment point clouds and the corresponding features from the previous stage are used for the query process with different radii and sample numbers. Then for each vertex, the pooled garment features are further encoded to aggregate low-level geometric features and high-level semantic information.

- **Hierarchical Body Surface Encoder** makes each vertex of the garment mesh aware of its neighbouring human body surface, which is important for correct interaction encoding between garment and human body, especially under the circumstances of large body movements. For each vertex of $M_p$, neighbouring human mesh vertices are sampled using different radii of query balls. Both coordinates and normal directions of sampled human mesh vertices are further encoded.

**Iterative GCN Displacement.** Concatenating the per-vertex features obtained from the above feature pooling and surface encoder, along with the coordinates of the vertex, we have aggregated necessary information for the following Iterative Graph Convolution Network (GCN) to predict the displacement $D$. Loose garments skinned by the interpolated LBS are normally not in accordance with the real situation. Therefore, we choose to iteratively predict displacements and gradually approach the ground truth. As shown in Fig. 3 (b), for the $j$-th iteration, using the previous accumulated displacement predictions $\sum_{i=1}^{j-1} D_i$ and the proposal $M_p$, we can calculate current mesh prediction by $M_{pj} = M_p + \sum_{i=1}^{j-1} D_i$. $M_{pj}$ is then used for the Proposal-Guided Hierarchical Feature Pooling of this iteration. The last GCN layer of each iteration outputs current displacement $D_j$. Inspired by [19], the GCN layer could be formulated as $H^{(l+1)} = \text{ReLU}(\bar{D}^{-1}\bar{A}H^{(l)}W^{(l)} + B^{(l)})$, where $\bar{A} = A + I$ is the bidirectional mesh graph with self-connections, $\bar{D} = \text{diag}(\sum_j \bar{A}_{ij})$, $W^{(l)}$ and $B^{(l)}$ are the learn-able weights and bias.

**Temporal Transformer.** To support better temporal information extraction, a Transformer is integrated into the Iterative GCN. For the $j$-th iteration, we take the features extracted by the $(j-1)$-th iteration, perform temporal fusion by Transformer and concatenate them to the input of the $j$-th iteration. Specifically, we denote the output feature of the second last layer of the $(j-1)$-th iteration as $F_{j-1} \in \mathbb{R}^{\mathcal{T} \times M \times K}$. Query, key and value are obtained by applying MLP to $F_{j-1}$, which could be denoted as $Q, K, V = \text{MLP}(F_{j-1})$, where $Q, K, V \in \mathbb{R}^{\mathcal{T} \times MK}$. Then the value is updated as $V' = \text{softmax}(QK^{\intercal}/\sqrt{T})V$. $V'$ is then concatenated to the pooled features of the $j$-th iteration.

## 2.5 Loss Functions

Loss functions corresponding to two parts of network are introduced in this section. For the canonical garment estimation, the loss function consists of five terms, which could be formulated as

$$
\begin{aligned}
\mathcal{L}_c =& \lambda_1 \mathcal{L}_{CE}(logits,\ label_{gt}) + \lambda_2 |\alpha' - \alpha_{gt}|_2^2 + \lambda_3 \sum_{i=1}^{M} \frac{1}{M} |T_i(\alpha') - T_i(\alpha_{gt})|_2^2 + \\
& \lambda_4 \mathcal{L}_{IL}\big(T_h, T(\alpha')\big) + \lambda_5 \mathcal{L}_{LR}\big(T(\alpha'), T(\alpha_{gt}), L\big).
\end{aligned}
\tag{4}
$$

The first term $\mathcal{L}_{CE}$ is the cross entropy term for semantic segmentation. The second and third term use L2 loss to supervise our predicted PCA coefficients $\alpha'$ and mesh vertices $T(\alpha')$, respectively. The fourth term is the interpenetration loss term, which could be formulated as

$$
\mathcal{L}_{IL}\big(T_h,\ T(\alpha')\big) = \sum_{i=1}^{M} \frac{1}{M} \text{ReLU}\Big( -\big(N_{hi} \cdot (T_i(\alpha') - V_{hi})\big)\Big),
\tag{5}
$$

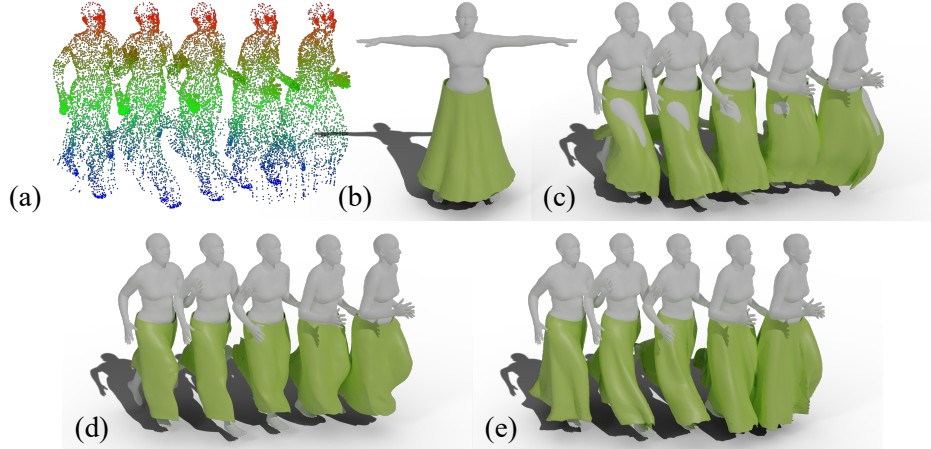

Figure 4: **Visualization of the Garment Reconstruction Process.** **(a)** Input point cloud sequence; **(b)** Canonical garment estimation; **(c)** Results of interpolated LBS; **(d)** Final reconstruction results; **(e)** Ground truth.

where $V_{hi}$ and $N_{hi}$ are the coordinate and normal vector of the nearest human mesh vertex of the $i$-th garment mesh vertex $T_i(\alpha')$. The fifth term is the Laplacian Regularization term,

$$\mathcal{L}_{LR}\big(T(\alpha'),\, T(\alpha_{gt}),\, L\big) = \sum_{i=1}^{M} \frac{1}{M} \Big( \big|LT(\alpha')\big|_2 - \big|LT(\alpha_{gt})\big|_2 \Big), \qquad (6)$$

where $L$ is the cotangent weighted Laplacian matrix of the garment mesh.

For the $j$-th iteration of the posed garment reconstruction, the loss function includes four terms, which are defined as

$$\mathcal{L}_{pj} = \lambda_6 \sum_{i=1}^{M} \frac{1}{M} \big|M' - M_{gt}\big|_2^2 + \lambda_7 \mathcal{L}_{IL}(M_h,\, M') + \lambda_8 \mathcal{L}_{LR}(M',\, M_{gt},\, L) + \lambda_9 \mathcal{L}_{TC}(M'). \quad (7)$$

The first term is the L2 loss term between predicted and ground truth garment mesh vertices $M'$ and $M_{gt}$. The second term penalizes interpenetration between posed garment prediction $M'$ and posed human body $M_h$. The third term performs cotangent weighted Laplacian regularization. The last term adds Temporal Constraints to the deformations of the predicted garments, which could be formulated as

$$\mathcal{L}_{TC}(M') = \sum_{t=2}^{T} \frac{1}{T-1} \big|M'(t) - M'(t-1)\big|_2^2. \qquad (8)$$

Adding up weighted loss of each iteration, we get the loss of the posed garment reconstruction $\mathcal{L}_p = \sum_j w_j \mathcal{L}_{pj}$.

## 3 Experiments

### 3.1 Datasets and Evaluation Protocols

**Datasets.** We establish point cloud sequence-based garment reconstruction dataset by adapting CLOTH3D [18] for our experiments. CLOTH3D [1] is a large scale synthetic dataset with rich garment shapes and styles and abundant human pose sequences. We sample point sets from 3D human models to produce the point cloud sequence inputs. We select three types of garments, *i.e.* Skirts, T-Shirt, Trousers, for the experiments. For skirts, 453 sequences with 109629 frames of point clouds are used for training and testing. For T-shirts, 394 sequences with 95540 frames are used. For trousers, 1465 sequences and 357086 frames are used. We split the sequences to training and testing sets at the ratio of $8 : 2$.

---

[1]CLOTH3D is downloaded from http://chalearnlap.cvc.uab.es/dataset/38/description/.

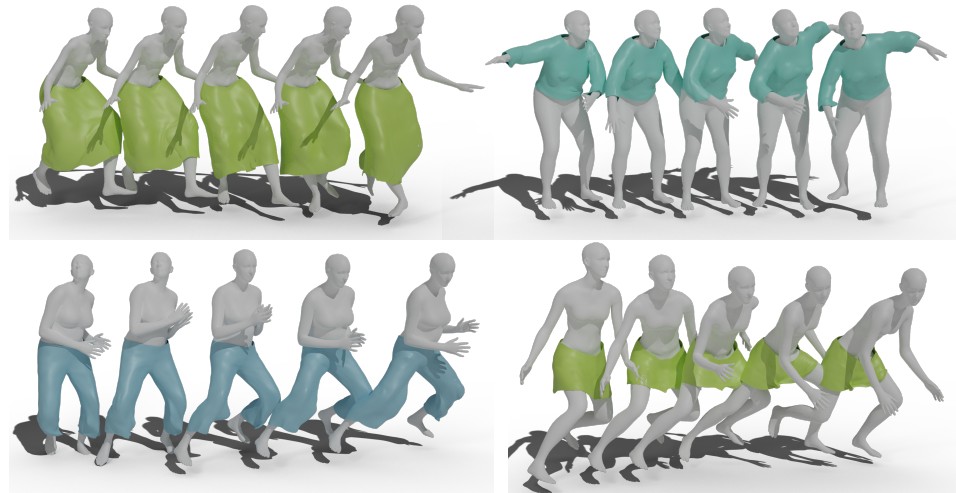

**Figure 5: Qualitative Reconstruction Results.** Reconstructed sequences of skirts, T-shirt and trousers are shown in different colors. High reconstruction quality is achieved even with large body movements.

| Class | PN++ | Ours |
|---|---|---|
| Skirt | 33.71 | **30.23** |
| T-Shirt | 41.05 | **28.39** |
| Trousers | 20.80 | **18.16** |

**Table 2:** L2 Error of Canonical Garment Estimation. PN++ refers to [21]. $[mm]$

| Class | MGN$^*$ | Int. LBS | Ours |
|---|---|---|---|
| Skirt | 71.66 | 79.52 | **49.39** |
| T-Shirt | 40.54 | 44.55 | **37.83** |
| Trousers | 29.12 | 31.05 | **26.48** |

**Table 3:** L2 Error of Posed Garment Reconstruction. MGN$^*$ is adapted from [16]. Int. LBS is the results of interpolated LBS. $[mm]$

| Class | MGN$^*$ | Ours |
|---|---|---|
| Skirt | 12.59 | **3.273** |
| T-Shirt | 7.173 | **4.749** |
| Trousers | 4.135 | **2.520** |

**Table 4:** Acceleration Error of Posed Garment Reconstruction. $[m/s^2]$

In addition to the synthetic dataset, we also perform experiments on a real human scan dataset CAPE [20]. CAPE [2] is a large-scale real clothed human scan dataset containing 15 subjects and 150k 3D scans. Since CAPE only releases the scanned clothed human sequences with no separable garment meshes, it is not practical to train our network on CAPE from scratch. Therefore, we directly inference on CAPE using the network pretrained on CLOTH3D.

**Evaluation Metrics.** Three metrics are utilized for evaluation on CLOTH3D. We first evaluate the per-vertex L2 error of canonical garments, $M_1 = \sum_{s,i} \frac{1}{MS} |T_{s,i}(\alpha') - T_{s,i}(\alpha_{gt})|_2$. Secondly, we evaluate the per-vertex L2 error of posed garments, which is defined as $M_2 = \sum_{f,i} \frac{1}{MF} |M'_{f,i} - M^{gt}_{f,i}|_2$. The last one is the acceleration error, which is used to evaluate the smoothness of the predicted sequences and could be formulated as $M_3 = \sum_{f,i} \frac{1}{MF} |A'_{f,i} - A^{gt}_{f,i}|_2$, where $A_{f,i}$ stands for acceleration vector of the $i$-th vertex of $f$-th frame.

Because the group truth garment meshes of CAPE are not provided, we evaluate the performance of different methods by using the 'one-way' Chamfer Distance from the vertices of the reconstructed meshes $R$ to the input clouds $P$. It is formulated as $\frac{1}{|R|} \sum_{r \in R} d_P(r)$, where $d_P(r)$ stands for the nearest distance from point $t$ to points in $P$.

**Comparison Methods.** Since it is the first work tackling garment reconstruction from point cloud sequences, there is no readily available prior work for direct comparison. Therefore, for canonical garment estimation, we adapt the PointNet++ [21] structure to predict the PCA coefficients directly from input point clouds. For posed garments reconstruction, we adapt Multi-Garment Net [16] (MGN) on point cloud inputs for comparison.

## 3.2 Qualitative Results

**Step-by-Step Visualization.** As shown in Fig. 4, we perform a step-by-step visualization of the whole reconstruction process. The estimated canonical skirt recovered rough shape and length of

---

$^2$CAPE is downloaded from https://cape.is.tue.mpg.de/.

the skirt from the point cloud sequences. As expected, the interpolation LBS produces reasonable proposals with smooth surfaces and no artifacts. There are two things that make the LBS results look unnatural. One is the interpenetration between garments and huamn body. The other is the lifted skirt floating in the air, not touching the leg, which should be the cause of the lift. These two flaws are inevitable because the skirt is not homotopic with human body. These defects would be eliminated finally. The final step of displacement prediction meets the expectations of three aspects. Firstly, the hierarchical garment feature pooling module captures the geometric features of cloth, which helps the precise shape recovery of posed garments. Secondly, the human body surface encoding module makes the network aware of the body surface positions, which prevents the interpenetration of garments and human body and helps in capturing the deformations caused by body movements, *e.g.* knees lifting up the skirt. Thirdly, the temporal fusion module encourages smooth deformation between consecutive frames.

**Reconstruction for Different Garment Types.** More visualization of the reconstruction results are illustrated in Fig. 5, which shows Garment4D's ability of handling both loose and tight garments driven by large body movements. For garments that are homotopic to human body, *i.e.* T-shirts and trousers, Garment4D can not only capture correct shape of them, but also the dynamics caused by body movement. Taking the bottom left trousers reconstruction as an example, when the right leg is stretched out, the back side of trousers stick to the back of calf. The front part of trousers naturally form the dynamics being pulled backwards. Moreover, it can be observed that when the leg or arm bends, more of the ankle or forearm are exposed in order to maintain the length of trousers legs or sleeves. For the reconstruction of skirts which are not homotopic to human body, both long and short skirts can be faithfully recovered. The short skirt is tighter and we can observe the natural deformation when the two legs are stretched to sides. For the looser long skirt, clear contacts and deformations can be observed when knees and calves lift. And when the legs are retracted to the neutral position, skirts would naturally fall back but would not stick to leg surfaces.

## 3.3 Quantitative Results on CLOTH3D

**Canonical Garment Estimation.** Tab. 2 reports the per-vertex L2 error of canonical garment estimation of the adapted PointNet++ [21] and our method. By explicitly making the network aware of the semantics our method out-performs the vanilla PointNet++ in all three garment types.

**Posed Garment Reconstruction.** The per-vertex L2 errors of posed garment reconstruction are shown in Tab. 3. Because T-shirt and trousers are homotopic to human body, the interpolated LBS step could give decent reconstruction results. In contrast, for skirts, The results of interpolated LBS of skirts is far from the real situation and therefore have high L2 errors, which supports the above qualitative analysis. The displacement prediction effectively captures the dynamics of skirts and improves $30.13\ mm$ on top of interpolated LBS. Moreover, Garment4D outperforms the adapted Multi-Garment Net [16] in all three garment types. Especially, Garment4D surpasses the adapted MGN in skirt by $22.27\ mm$, which further shows the advantage of Garment4D over SMPL+D-based methods at reconstructing loose garments like skirts.

**Temporal Smoothness.** As reported in Tab. 4, our method also outperforms the adapted MGN in terms of reconstruction smoothness. Even though the same temporal constraints term is included in the loss functions of both methods, the adapted MGN fails to properly model the motion of garments and aggregate temporal information. Particularly for loose garments like skirts, Garment4D outperforms the adapted MGN by $9.317\ m/s^2$.

| Class | MGN$^*$ | Ours |
|---|---|---|
| T-Shirt | 0.430 | **0.366** |
| Trousers | 0.922 | **0.455** |

**Table 5:** 'One-way' Chamfer Distance on CAPE [$mm$]. MGC$^*$ is adapted from [16].

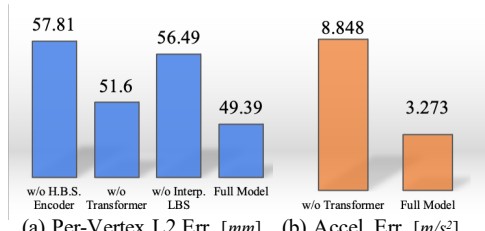

(a) Per-Vertex L2 Err. [*mm*]    (b) Accel. Err. [*m/s²*]

**Figure 6: Results of Ablation Study.** (a) shows the comparison of Per-vertex L2 error of posed garment reconstruction. (b) reports the acceleration error.

### 3.4 Quantitative Results on CAPE

As reported in Tab. 5, our method outperforms the adapted MGN on both classes in terms of the 'one-way' Chamfer Distance. It shows the effectiveness and advantages of Garment4D on real world data. Moreover, we directly inference on CAPE with the model trained on CLOTH3D without fine tuning, which further demonstrated the generalizability of Garment4D.

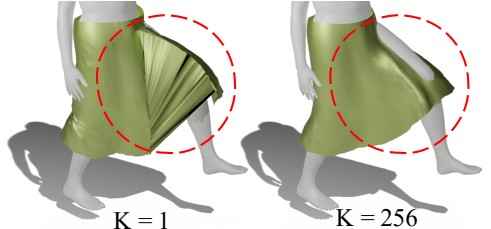

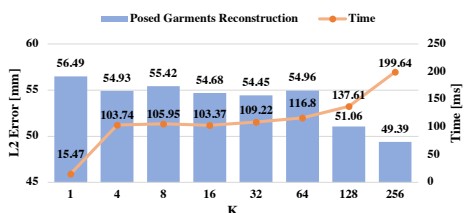

**Figure 7: Comparison of LBS results when nearest neighbour** $K$ **is set to** $1$ **and** $256$**.** Severe artifacts appear when $K = 1$, which is circled out.

**Figure 8:** Trade-off between number of neighbors $K$, the reconstruction performance and computational cost.

### 3.5 Ablation Study

**Hierarchical Body Surface Encoder.** As shown in Fig. 6, without the hierarchical body surface encoder (H.B.S Encoder), the L2 error increases by $8.42\ mm$, which is in line with the expectations. Without knowing body positions, it is hard for networks to infer garment-body interactions.

**Temporal Transformer.** As illustrated in Fig. 6, if we remove the temporal transformer, the L2 error increases by $2.21\ mm$ and the acceleration error increases by $5.575\ m/s^2$. The results indicate that the temporal transformer not only helps the single frame reconstruction by aggregating information from neighbouring frames, but also increases the smoothness of the reconstruction sequences.

**Interpolated LBS.** As circled out in the left side of Fig. 7, if the skinning weights of garment vertices are copied from the nearest body vertex, which is equivalent of SMPL+D (*e.g.* MGN[16]), the severe cliff-like artifacts appear between the legs. The quality of the LBS proposals would effect the final reconstruction results, which is shown in Fig. 6. If $K = 1$, the per-vertex L2 error of the final reconstruction results would increase by $7.1\ mm$.

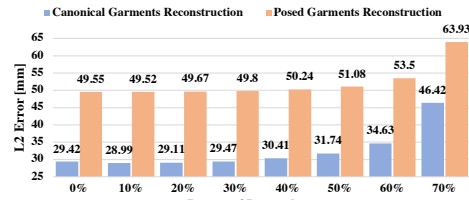

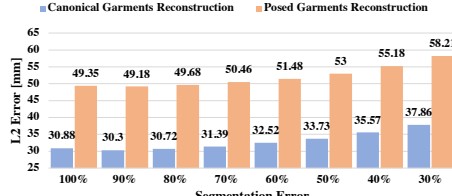

**Figure 9:** Reconstruction Performance on Point Clouds with different levels of incompleteness.

**Figure 10:** Reconstruction Performance given Different Segmentation Errors.

### 3.6 Further Analysis

**Number of Neighbors** $K$ **of Interpolated LBS.** As shown in Fig. 8, the performance peaks at $K = 256$, which explains our parameter choice. The K nearest neighbors and weights interpolation are implemented for GPU parallel computation. Therefore, the computation overhead is minor with the increase of $K$.

**Performance on Incomplete Point Clouds.** The real-scanned point clouds can be incomplete, which makes an important setting to test on. To control the degree of incompleteness (*i.e.* the percentage of missing points), we randomly crop out holes from the input point clouds and performs evaluation on the skirt test set of CLOTH3D. As shown in Fig. 9, decent reconstruction results can be obtained at up to 50% of incompleteness, which shows the robustness of our methods on imperfect point clouds.

**Robustness of Reconstruction Results over Segmentation Errors.** The segmentation quality of input point clouds can affect the reconstruction results. To investigate such effects, we construct

different degrees of segmentation errors and evaluate on the skirt test set of CLOTH3D. As shown in Fig. 10, a decrease of reconstruction quality can be observed with the decrease of segmentation accuracy. However, even with the accuracy of 50%, our method can still produce decent reconstruction results. Therefore, Garment4D is robust to segmentation errors.

## 4 Related Work

**Template-Free Clothed Human Modeling.** Most previous template-free methods reconstruct clothed human use volumetric representation or implicit function. [4, 5, 9–13] reconstruct clothed human from images using volumetric representation. Due to large memory footprint of such representation, these methods usually have trouble in recovering fine details of surfaces. [6, 22–25] propose efficient clothed human reconstruction pipeline with the help of implicit function. The above two types of methods both reconstruct human and cloth as a whole, which means it is not possible to directly take higher level of control of the garments, *e.g.* re-pose, re-targeting.

**Template-Free Garment Modeling.** In order to focus more on the garments, several recent work make some attempts to reconstruct human and garments separately using implicit function. [15] predicts a layered implicit function, which makes it possible to separate garments from human body. [14] builds a garment specific implicit function over the canonical-posed human body, which could be controlled by latent codes that describe garment cut and style. Not relying on any specific template could be an advantage (*e.g.* more degrees of freedom). But in some applications *e.g.* re-posing, re-targeting, texture mapping, it is a disadvantage because of poor interpretability. To achieve higher level of control, [15, 14] have to register the garment meshes, which are reconstructed from implicit functions, to the SMPL+D model.

**Parametric Clothed Human Modeling.** Most previous garment reconstruction work that use parametric methods tend to take human body and garment as a whole. A common practice, SMPL+D [7, 8, 26–30], is to predict displacements from SMPL [31] vertices. However, SMPL+D has difficulty modeling loose garments or garments that do not share same topology as human body, which limits the expression ability of SMPL+D.

**Parametric Garment Modeling.** In order to reconstruct garments separately using parametric models, PCA decomposition is used by [16, 17] to model the single layered garment meshes, which is independent of any parametric human body model. To further increase the expressive ability of an individual parametric garment model, [32] maps garments to two types of UV-maps according to whether the garment is homotopy to human body. As for the deformation of the garment models, most existing methods rely on the skinning process of human body. [16] samples the skinning weights of garment vertices from the closest human body vertices [17] predicts skinning weights of garment vertices using neural network. Only relying on the skinning process would lead to overly smoothed results, which is addressed by [33] through additional displacement prediction before skinning.

## 5 Conclusion

In this work, we present the first attempt at garment reconstruction from point cloud sequences. A garment model, with the novel interpolated LBS, is introduced to adapt to loose garments. A garment mesh registration method is proposed to prepare sequential meshes for the garment reconstruction network. Taking point clouds sequences as input, the corresponding canonical garment is estimated by predicting PCA coefficients. Then a Proposal-Guided Hierarchical Feature Network is introduced to perform semantic, geometric and human-body-aware feature aggregation. Efficient Iterative GCN along with Temporal Transformer is utilized to further encode the features and predict displacements in an iterative way. Qualitative and quantitative results from extensive experiments demonstrate the effectiveness of the proposed framework.

**Acknowledgments** This study is supported by NTU NAP, MOE AcRF Tier 1 (2021-T1-001-088), and under the RIE2020 Industry Alignment Fund – Industry Collaboration Projects (IAF-ICP) Funding Initiative, as well as cash and in-kind contribution from the industry partner(s).

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
