# OpenReview forum: "Garment4D: Garment Reconstruction from Point Cloud Sequences"
_NeurIPS.cc/2021/Conference — NeurIPS 2021 Poster_

### Official Review · Reviewer_cJ86 · 2021-07-12

**Rating:** 7
**Confidence:** 3

**Summary:**

This paper proposed a framework, Garment4D, to reconstruct 3D garment models from 3D point cloud sequences of dressed humans instead of 2D RGB images. There are three key components in the proposed framework, including garment registration, garment shape estimation in the canonical pose, and garment reconstruction in the target pose. Experiments demonstrate the effectiveness of the proposed framework with more accurate and natural reconstruction results.

**Limitations And Societal Impact:**

Reconstructing 3D garment models from 3D point cloud sequences is interesting and appealing. However, this submission does not show or disccuss the performance on real-world point cloud sequences. In comparsion of the solutions using 2D images as inputs, the generazation of the proposed framework remains unclear.

**Main Review:**

This submission claims itself as the first attempt to reconstruct the 3D garment model from 3D point cloud sequences, which eliminates many limitations of those solutions using 2D images. The three key steps in the proposed framework also include new technical designs to address the challenges in modeling the detailed deformation and garment dynamics. Overall, this paper is well-written and technically sound.

There are a few concerns regarding this paper:

- Guo et al. have also investigated using point cloud as input for the reconstruction of dynamic geometry of clothes. Though it seems that these two papers are concurrent with each other, it is recommended to add discussions or comparisons with Guo et al. in the final version of the submission.
[Guo et al., Inverse Simulation: Reconstructing Dynamic Geometry of Clothed Humans via Optimal Control, In CVPR, 2021]

- In Sec. 2.3, point-wise semantic segmentation is needed to select the garments from the input point cloud. In practice, it may be harder for algorithms to perform clean segmentation from real-world point clouds. How about the robustness of the estimation results over the segmentation errors?

- In Sec. 2.4 and Fig. 3, are all hierarchical garment and body features concatenated together as the input of GCN? It is not very clear in the presentation and figure since the vertex numbers of the garment and body are different.

- In Sec. 3.4 and Fig. 6, how about the trade-off between the K value and the reconstruction performance? How about the additional computational cost raising by the increase of K?

**Time Spent Reviewing:**

8

---

> ### Author Response · Authors · 2021-08-10
> **Response to Reviewer cJ86**
>
> Thank you for your acknowledgment of our effort and for the valued advice.
>
> ### Q1: Compare to Guo et. al.
> Thank you for bringing this recent work to our attention. Guo et. al. [1] incorporate physical simulation into the reconstruction process, which is an interesting and effective way towards the goal. We would add the discussion and cite the paper in the final revision.
>
> ### Q2: The robustness of the estimation results over the segmentation errors.
> Thank you for asking the important question. It is true that the segmentation quality does affect the final reconstruction quality. To better control the segmentation quality, we manually construct different degrees of segmentation quality and evaluate on the skirt test set of Cloth3D. To be more specific, we randomly sample mini clusters from the skirt point clouds and gives them wrong labels. Then we randomly pick non-skirt points and assign them the skirt label to introduce noisy segments. During inference time, instead of using the segmentation predicted by the network, we use the manually constructed imperfect segmentation as the input to the following modules. Just to be clear, when training the network, we do not manually construct noisy segmentation as augmentation. We report the results in the table below. We observe a decrease in reconstruction quality with the decrease of segmentation accuracy, which is in line with our expectations. Even with an accuracy of 50%, our method can still produce decent reconstruction results. Therefore, it is safe to say that our method is robust to segmentation results. We will add the experiments into the discussion of the main paper.
>
> | Segmentation Accuracy | 100%  |  90%  |  80%  |  70%  |  60%  |  50%  |  40%  |  30%  |
> | :-------------------- | :---: | :---: | :---: | :---: | :---: | :---: | :---: | :---: |
> | L2 Error [mm]         | 49.35 | 49.18 | 49.68 | 50.46 | 51.48 | 53.00 | 55.18 | 58.21 |
>
> ### Q3: Not clear about the input of the GCN.
> Your understanding is correct, and all the features from hierarchical garment feature pooling, hierarchical body surface encoding, temporal transformer, and the coordinates of vertices are concatenated to input into the GCN. Although the number of vertices of the body and garments is different, the features here are all pooled to each vertex of the garment mesh. Therefore they have the same shape and hence can be concatenated together.
>
> ### Q4:  Trade-off between K and the reconstruction performance and additional computational cost.
> Thank you for asking the question. We take your advice and conduct experiments about the choice of K and its effects on the performance of the network. As shown in the table below, the performance peaks at K=256, which explains our parameter choice. Due to the GPU memory limitation, we could not further increase K. The K nearest neighbors and weights interpolation are implemented for GPU parallel computation. Therefore, the time overhead is minor with the increase of K.
>
> | K             |   1   |   4   |   8   |   16  |   32  |   64  |  128  |  256  |
> | :------------ | :---: | :---: | :---: | :---: | :---: | :---: | :---: | :---: |
> | L2 Error [mm] | 56.49 | 54.93 | 55.68 | 54.87 | 54.68 | 54.96 | 51.06 | 49.39 |
> | Time [ms]     |  15.5 | 103.74| 105.95| 103.37| 109.22| 116.80| 137.61| 199.64|
>
> ### Q5:  The performance on real-world point cloud sequences and the generalization of the proposed framework.
> Please kindly refer to 'Response to the Common Request of Experiments on Real Data' for the real data experiments. Considering the fact that we directly use the model trained on synthetic data (Cloth3D) to inference on real data (CAPE), it is fair to say that our method has a certain ability to generalize.
>
> #### References
> [1] Jingfan Guo, Jie Li, Rahul Narain, and Hyun Soo Park. Inverse simulation: Reconstructing dynamic geometry of clothed humans via optimal control. In Proceedings of the IEEE/CVF Conference on Computer Vision and Pattern Recognition, pages 14698–14707, 2021.

---

> > ### Author Response · Authors · 2021-08-18
> > **Follow-up**
> >
> > Dear reviewer,
> >
> > We have provided further analysis of our method and addressed your concerns about the generalizability of the proposed framework by providing the real data experiment. We would love to hear your feedback on whether our answers have solved your concern or if you have further questions.

---

> > > ### Comment · Reviewer_cJ86 · 2021-09-01
> > > **Reply to author's response**
> > >
> > > Thanks for the rebuttal. The new results have addressed most of my concerns, so I will keep my original rating. One more question is: will the code be publicly released? The code should greatly facilitate the research of this research area.

---

> > > > ### Author Response · Authors · 2021-09-01
> > > > **Thank you for the feedback**
> > > >
> > > > Thank you for the feedback. And yes, we will release the code upon paper acceptance.

---

### Official Review · Reviewer_1Hss · 2021-07-16

**Rating:** 6
**Confidence:** 5

**Summary:**

This manuscript designs a parametric garment model and utilize techniques such as point cloud network and GCN to reconstruct garment mesh. Quantitative and qualitative experiments on synthetic data show the effectiveness of this method to a certain extent. Compared to MGN [16], the method can better process loose garments and produce more natural reconstructions. The technical description is also clear and reproducible.

**Ethics Review Area:**

["I don’t know"]

**Limitations And Societal Impact:**

As discussed above, the authors should analyze more about the practicality and scalability of the method.

**Main Review:**

I have several concerns:
1. Its parametric garment representation, including the interpolated LBS skinning weights computation, is very similar with the garment model of BCNet[17], except it adds displacements on posed garments. I think this will decrease the manuscript's novelty.
2. The authors claim that they are the first to explore garment reconstruction from point cloud sequences. It may be true while I do not think it is very importmant. First, some works have explore the clothed body reconstruction from point clouds [A,B], while they use implicit representations. Second, point cloud input can bring other problems. For example, real point cloud data often has holes and may be partial, these can greatly affect the prediction of the network, while these situations are not analysized in the manuscript.
3. If I understand correctly, the method not only needs the garment point cloud, but also the dressed body point cloud, which is used in the Hierarchical Body Surface Encoding. I think this requirement severely restricts the application of this method to real scene. Because in a realistic scanning environment, it is difficulty to get the body scanning hidden under the garments. Therefore, I doubt the practicality of this method.
4. All the experiments are based on the synthetic data, while some real point cloud should be tested to present the practicality. Besides, the demo presented reconstruction results are not satisfactory  for me.

[A] Saito S, Yang J, Ma Q, et al. SCANimate: Weakly supervised learning of skinned clothed avatar networks[C]//Proceedings of the IEEE/CVF Conference on Computer Vision and Pattern Recognition. 2021: 2886-2897.
[B] Chen X, Zheng Y, Black M J, et al. SNARF: Differentiable Forward Skinning for Animating Non-Rigid Neural Implicit Shapes[J]. arXiv preprint arXiv:2104.03953, 2021.

**Time Spent Reviewing:**

3.5

---

> ### Author Response · Authors · 2021-08-10
> **Response to Reviewer 1Hss**
>
> Thank you for your helpful comments.
>
> ### Q1: Similar to the garment model of BCNet.
> Our apologies for the possible disorientation caused, but we disagree that our parametric garment model is similar to that of BCNet [1]. Two main differences between ours and that of BCNet are given below.
> 1. Our interpolated LBS does not require to be learned, while BCNet has to use a neural network to learn the LBS weights.
> 2. In our garment model, we use the interpolated LBS to provide a reasonable reconstruction proposal. Following, we predict displacements to further improve the reconstructed details and dynamics. Unlike our method, BCNet directly uses the LBS results as their final results.
>
> Due to the two aforementioned differences, our garment model is more suitable for sequential garment dynamics modeling than BCNet. The major reason is that the LBS weights of BCNet are only conditioned on garment shapes and body poses, and it totally overlooks movements and dynamics. In other words, given the same body pose, BCNet will predict the same reconstructed garments. However, given the same body pose, the garments can actually have far different reconstructions caused by different sequential movements (e.g. walking or running). Rather than only considering single-frame garment reconstruction, our method can resolve sequential garment reconstruction by the 'proposal and displacement' paradigm. Specifically, the predicted displacements can adaptively capture sequential movements and dynamics to generate better garment reconstructions. We will add the above discussions in the revised version to highlight our novelty and advantages.
>
> ### Q2:  Garment reconstruction from point cloud sequences is not very important.
> We beg to differ with your opinion on the importance of the novel task, garment reconstruction from point cloud sequences.
>
> Firstly, we think it is an important task to explore reconstruction from point clouds inputs. Academically, such a task poses great opportunities to apply and validate theories in geometric deep learning and point clouds representation learning, which would encourage advancement in these areas. Practically, it is becoming a recent trend to learn geometric features directly from 3D point clouds for the following three main reasons: 1) raising concerns of a possible privacy leak from images, especially for human-related tasks; 2) more and more available 3D sensors; 3) that 3D inputs could provide accurate scale while images could not. We agree that real-scanned point clouds can be partial, noisy, and imperfect, which makes it a challenging problem. However, as the first work, we believe it is a valuable effort to show the effectiveness of geometric features learning from point cloud sequences on dynamic 3D reconstruction. We hope this work can inspire many future research works on garments reconstruction based on 3D point clouds.
>
> Secondly, we emphasize that there are many advantages of parametric garment reconstruction compared to implicit reconstruction. As discussed in the first two paragraphs of the main paper, 1) the garments are normally not separable from the reconstructed implicit representation of the clothed human; 2) one more step of marching cube is required to produce the render-ready meshes; 3) the implicit representation or the meshes obtained from the marching cube lacks interpretability, or in other words, it could not provide dense correspondences. The ability of parametric garment models to reconstruct the interpretable garment meshes is desirable for related industries.
>
> To further relieve your concerns on imperfect point cloud scans, we provide additional experiments to show the performance of our method when facing incomplete point clouds with holes. To control the degree of incompleteness (i.e. the percentage of discarded points), we randomly crop out holes from the input point clouds and performs evaluation on the skirt test set from Cloth3D [2]. As shown in the table below, decent reconstruction results can be obtained at up to 50% of incompleteness. We will add the experiments into the discussion of the main paper. As for real-scanned partial point clouds (2.5D), it is out of the scope of this paper, as we mainly focus on parametric garments reconstruction from full point clouds (3D). As for the feasibility of obtaining full point clouds, it is guaranteed by full-body 3D scanners like 3dMD LLC, which is used by CAPE [3] to capture full scans of sequential clothed humans.
>
> | Degree of Incompleteness |   0%  |  10%  |  20%  |  30%  |  40%  |  50%  |  60%  |  70%  |
> | :----------------------- | :---: | :---: | :---: | :---: | :---: | :---: | :---: | :---: |
> | L2 Error [mm]            | 49.55 | 49.52 | 49.67 | 49.80 | 50.24 | 51.08 | 53.50 | 63.93 |
>
> In summary, we think it is an important and valuable research topic to reconstruct parametric garment models from 3D point cloud sequences. Moreover, our method shows robustness in dealing with imperfect point clouds.
>
> ### Q3: The dressed body point clouds are needed.
> Our method requires human body surfaces, but it will not 'severely restrict the application of this method to the real scene'. As shown in the real data experiments (CAPE), we use the SMPL model to provide body surfaces, which are estimated by optimization based on input point clouds. In particular, the better performance achieved by our method than the adapted MGN [4] is reported. Therefore, we argue that our work is able to provide a feasible solution for real-world applications.
>
> ### Q4: Real data should be tested.
> Please kindly refer to 'Response to the Common Request of Experiments on Real Data'.
>
> ### Q5: The reconstruction results are not satisfactory for me.
> We present the first effort to reconstruct loose garments from sequential point cloud inputs in a parametric way, which is not achievable in previous works. Of course, there remains a lot of work before we can obtain perfect reconstruction results. But as the first work in this novel task, we construct a novel method that is able to produce much better results than adapted baselines. And thorough experiments validate the effectiveness of each part of the design.
>
> #### References
> [1] Boyi Jiang, Juyong Zhang, Yang Hong, Jinhao Luo, Ligang Liu, and Hujun Bao.  Bcnet: Learning body and cloth shape from a single image. In European Conference on Computer Vision, pages 18–35. Springer, 2020.
>
> [2] Hugo  Bertiche,  Meysam  Madadi,  and  Sergio  Escalera. Cloth3d: Clothed 3d  humans. In European Conference on Computer Vision, pages 344–359. Springer, 2020.
>
> [3] Qianli Ma, Jinlong Yang, Anurag Ranjan, Sergi Pujades, Gerard Pons-Moll, Siyu Tang, and Michael J Black. Learning to dress 3d people in generative clothing. In Proceedings of the IEEE/CVF Conference on Computer Vision and Pattern Recognition, pages 6469–6478, 2020.
>
> [4] Bharat Lal Bhatnagar, Garvita Tiwari, Christian Theobalt, and Gerard Pons-Moll. Multi-garment net: Learning to dress 3d people from images. In Proceedings of the IEEE/CVF International Conference on Computer Vision, pages 5420–5430, 2019.

---

> > ### Author Response · Authors · 2021-08-18
> > **Follow-up**
> >
> > Dear reviewer,
> >
> > We have clarified the difference between our method and BCNet, re-emphasized the importance of the proposed task, and addressed your concerns about the practicality of our method. We would love to hear your feedback on whether our answers have solved your concern or if you have further questions.

---

> > ### Comment · Reviewer_1Hss · 2021-08-26
> > **Reply to author's response**
> >
> > First, thanks to the authors for their efforts to clarify my concerns, and I will to increase my score after reading the rebuttal. My main concern is about the real point cloud input. The supplementary experiment on CAPE has confirmed its superiority over adapted MGN baseline for real 3D point clouds. On the other hand, I do not think garment reconstruction from point cloud sequences is not important, but some discussions should be given to common partial point cloud input. The incompleteness experiment on Cloth3D has clear my concern on this point to a certain extent. Moreover, the CAPE experiment also presents the feasibility of estimating SMPL shape from real 3D point clouds, while utilizing gt poses. I think it will be very interesting to analyze the performance based on some automatic SMPL estimation methods. However, the proposed differences with BCNet, 'BCNet directly uses the LBS results as their final results' is not correct. BCNet also predicts displacements to refine details, while it adds the displacements before LBS. Authors should revise the discussions.

---

> > > ### Author Response · Authors · 2021-08-26
> > > **Thank you for raising the rating.**
> > >
> > > Thank you for your feedback and for raising the rating. We agree that it is an interesting and open question to reconstruct both SMPL and garments from point clouds, considering that estimating only SMPL parameters from point clouds is a recently emerging task. Moreover, we agree that BCNet predicts displacements before the LBS. We will further revise the related discussion. Overall, thank you for your constructive advice. We will add the discussions to the revised version.

---

### Official Review · Reviewer_jo5B · 2021-07-19

**Rating:** 7
**Confidence:** 4

**Summary:**

The paper presents an approach for 4D garment reconstruction. To this end, it first presents a statistical shape model for garments which is learned using sequences of garment meshes. Since garment meshes across different sequences may have different topologies, the paper aligns them by first registering them to a template mesh and then re-meshing using barycentric interpolation. Given the statistical garment model, the paper then presents an approach for 4D garment reconstruction from a point-cloud sequence as input. The point cloud is first segmented into different body parts and then fed into an MLP to predict the PCA coefficients of the garment model. This provides the garment mesh in the canonical pose. Subsequently, for each body pose, the canonical mesh is posed using the LBS skinning weights interpolated from the body model. These posed meshes are further refined by iteratively predicting additional displacements using hierarchical feature extractors and a transformers-based temporal model. Experiments are performed on the CLOTH3D dataset and compared with adapted baselines. The proposed method is shown to outperform the baselines. Additional ablative experiments demonstrate the importance of different design choices.

**Ethical Concerns:**

No ethical concerns.

**Limitations And Societal Impact:**

No limitations and societal impacts are provided. I suggest authors to kindly address this by providing results on faster motions, different body shapes, and providing additional comments about societal impacts.

**Main Review:**

**Pros:**
- The paper addresses the problem of garment reconstruction from point-cloud sequences which has not been addressed previously and is of great interest, though I am not sure if the paper aligns well with NeurIPs (may be better suited to CVPR, etc.)
- The proposed approach is intuitive and makes sense, although it's incremental in nature.
- Appropriate baselines are developed to validate the contributions of the paper.
- Sufficient experiments are performed to validate the design choices.
- I like the idea of interpolated LBS to avoid cliff-like skinning artifacts.
- I like the idea of using interpenetration loss to explicitly build a relationship between body surface and clothing.

**Cons:**
- No qualitative samples for real data are provided.
- Also, no examples of very fast body movements is provided in the supplemental video. I mainly wanted to evaluate the impact of laplacian smoothening. For slow motion, the resulting reconstruction looks good, but how does it look for fast motion? Is there any lag in general?
- The paper is a bit hard to follow at times. I sometimes missed the intuitions behind different design choices which makes the paper less interesting to read. Also, the paper requires a thorough revision. I have provided some typos below.
- In Section 3.4, when replacing a specific component of the proposed approach, it's not clear what is used instead. Please make it clear by also providing the details of alternative components.
- The related work is not enough and requires a thorough revision.

**Questions:**
- For the BC term in eq (6), how are the corresponding boundaries obtained?
- How are the human body meshes obtained? Are ground-truth meshes used in this case?

**Missing References:**
The following papers should be referenced in the paper as they also tackle estimation of clothing:
1. MonoClothCap: Towards temporally coherent clothing capture from monocular RGB video, 3DV'20.
2. DeepCap: Monocular Human Performance Capture Using Weak Supervision, CVPR'20.

Also, 27 references are too few to do justice with the large amount of literature on human clothes modeling. Please thoroughly revise the related work.

**Typos:**
L67: learns -> learn
L76: are formulated -> is formulated.
L77: afterwards and successively are redundant.
L87: re presents -> represents
L117: normalized -> normalizes
L174: Biases
L228: homotopy -> homotopic
L229: of all three aspects.
L334: and help in capturing the deformations caused by body movements.

**Other suggestions:**
The work canonical-posed can be replaced with `Canonical` for ease of reading. This, however, is a personal preference and not critical.

**Time Spent Reviewing:**

5

---

> ### Author Response · Authors · 2021-08-10
> **Response to Reviewer jo5B**
>
> Thank you for your acknowledgment of our work and for your valuable advice.
>
> ### Q1: No qualitative samples for real data are provided.
> Please kindly refer to 'Response to the Common Request of Experiments on Real Data'.
>
> ### Q2: Very fast body movement.
> We actually include fast body movements (e.g. kicking legs and running) in our demo video, and we will highlight those cases in the revised version. All motion sequences in the demo video are originally in 30 fps, which are slowed down to 15 fps in order to better show reconstruction details. Empirically, the laplacian smoothening of one frame would not cause obvious lags in sequences. Moreover, we randomly skip some frames in input sequences as a training time augmentation, which makes our network able to adapt to different motion speeds, especially for fast body movements.
>
> ### Q3: Need thorough revision.
> Thank you. We will correct all the typos accordingly, and will also improve the understandability readability of our paper in the revised version.
>
> ### Q4: In Sec. 3.4, the details of alternative components are not clear.
> 1. For the ablation study on the Hierarchical Body Surface Encoder, we remove the encoder, which means the features input into the GCN are the features from 1) Hierarchical Garment Feature Pooling module, 2) Temporal Transformer and 3) the coordinates of the vertices.
> 2. For the ablation study on the Temporal Transformer, we remove the transformer and directly concatenate the features from the hidden layers of the last iteration to the input features of the next iteration.
> 3. For the ablation study on the Interpolated LBS, we set the nearest neighbors K to 1 so that it degenerates to the conventional LBS that MGN [1] and BCNet [2] use.
>
> We will update the above details to the main paper in the revised version.
>
> ### Q5: The related work is not enough.
> Thank you for the advice. Due to the space limitation, we discussed many related works in our supplemental material. We will revise and provide a thorough discussion on related works in the main paper.
>
> ### Q6: For the BC Term in eq (6), how are the corresponding boundaries obtained?
> Thank you for asking. I think you mean the BC term in eq (4). First, we find the boundary edges in the garment meshes using a simple observation that each boundary edge only belongs to one face, while each inside edge belongs to two faces. After finding the boundary edges, we further construct boundary loops by searching through the graph provided by the mesh. Finally, since the garments are in the canonical pose, it is straightforward to find the corresponding boundary loops by using the relative positions.
>
> ### Q7: How are the human body meshes obtained?  Are ground-truth meshes used in this case?
> In the experiments on Cloth3D [3], we use the ground truth SMPL meshes as the human body meshes. For real data (e.g. the CAPE experiment explained above), we could estimate SMPL parameters based on the input point clouds and use the estimated SMPL mesh.
>
> ### Q8: Other suggestions on limitation and societal impact analysis.
> Thank you for the suggestions. Cloth3D [3] is a large-scale synthetic dataset that contains a wide variety of body shapes. We will add qualitative results of different body shapes in the final revision.
>
> We believe that the possible societal impacts are in two ways. Firstly, as a human-centric work, diversity in gender and ethnic should be considered. Cloth3D uses SMPL as the human body model to synthesize data. While the gender is balanced, further efforts are required to further take ethnic diversity into consideration. Secondly, privacy is a raising social issue in recent years. Different from images, using point clouds minimizes the danger of privacy leaks, which is a positive contribution of our work.
>
> #### References
> [1] Bharat Lal Bhatnagar, Garvita Tiwari, Christian Theobalt, and Gerard Pons-Moll. Multi-garment net: Learning to dress 3d people from images. In Proceedings of the IEEE/CVF International Conference on Computer Vision, pages 5420–5430, 2019.
>
> [2] Boyi Jiang, Juyong Zhang, Yang Hong, Jinhao Luo, Ligang Liu, and Hujun Bao.  Bcnet: Learning body and cloth shape from a single image. In European Conference on Computer Vision, pages 18–35. Springer, 2020.
>
> [3] Hugo  Bertiche,  Meysam  Madadi,  and  Sergio  Escalera. Cloth3d: Clothed 3d  humans. In European Conference on Computer Vision, pages 344–359. Springer, 2020.

---

> > ### Author Response · Authors · 2021-08-18
> > **Follow-up**
> >
> > Dear reviewer,
> >
> > We have provided the real data experiment to show the generalizability of our method. And we have clarified a few technical details you mentioned. We would love to hear your feedback on whether our answers have solved your concern or if you have further questions.

---

> > > ### Comment · Reviewer_jo5B · 2021-09-01
> > > **Follow-up**
> > >
> > > Dear Authors,
> > >
> > > Thanks a lot for the follow-up. I have read the rebuttal and don't have any other questions. Thank you!

---

### Author Response · Authors · 2021-08-10
**Response to the Common Request of Experiments on Real Data**

Thank you all for the suggestion. We choose the publicly available real human scan dataset CAPE [1] to test our method. CAPE is a large-scale real clothed human scan dataset that contains 15 subjects and 150k 3D scan frames. Each subject wears long or short sleeve shirts and trousers. Since CAPE only releases the scanned clothed human sequences with no separable garment meshes, it is not practical to train our network on CAPE from scratch. Therefore, we use the network pretrained on Cloth3D and then directly inference on CAPE without fine-tuning. CAPE only provides the human pose for each frame. Therefore we have to estimate the SMPL [2] body shape parameters by optimization. The estimated SMPL meshes are used as the human body surface inputs. We report the reconstruction results of both trousers and shirts on all 15 subjects. Because the ground truth garment meshes are not provided, we evaluate performance of different methods by using the 'one-way' Chamfer Distance from the vertices of the reconstructed meshes $R$ to the input point clouds $P$, which can be formulated as $\sum_{r\in R}d_{P}(r)$, where $d_{P}(t)$ stands for the nearest distance from point $t$ to any point in $P$. In order for better understanding of the scale of the numbers, for the reconstructed trousers mesh $R_{t}$, $|R_{t}| = 8095$. While for the reconstructed shirts mesh $R_{s}$, $|R_{s}| = 6200$. We also report the results of the pretrained adapted MGN [3] as a fair comparison. The first table below shows the reconstruction results of trousers and the second one shows the results of shirts. Our method consistently outperforms the adapted MGN in all subjects. The qualitative results by our method, which unfortunately cannot be shared due to the OpenReview system limitation, look reasonable and impressive, and hence our method is able to capture the details and dynamics of the sequential scans for real data. We will add the quantitative and qualitative results of the real dataset in the final version.

| Subject     | 00032 | 00096 | 00122 | 00127 | 00134 | 00145 | 00159 | 00215 | 02474 | 03223 | 03284 | 03331 | 03375 | 03383 | 03394 |
| :------     | :---: | :---: | :---: | :---: | :---: | :---: | :---: | :---: | :---: | :---: | :---: | :---: | :---: | :---: | :---: |
| Adapted MGN | 6.26  | 6.88  | 5.67  | 5.61  | 6.43  | 6.85  | 5.56  | 7.31  | 12.3  | 6.09  | 6.58  | 4.89  | 9.23  | 6.70  | 9.79  |
| Ours        | 3.08  | 2.91  | 2.67  | 2.64  | 3.78  | 3.37  | 2.81  | 3.13  | 5.70  | 2.97  | 2.95  | 2.70  | 4.58  | 3.04  | 4.76  |

| Subject     | 00032 | 00096 | 00122 | 00127 | 00134 | 00145 | 00159 | 00215 | 02474 | 03223 | 03284 | 03331 | 03375 | 03383 | 03394 |
| :------     | :---: | :---: | :---: | :---: | :---: | :---: | :---: | :---: | :---: | :---: | :---: | :---: | :---: | :---: | :---: |
| Adapted MGN | 2.82  | 2.69  | 2.32  | 2.71  | 2.46  | 2.58  | 2.18  | 2.49  | 3.68  | 2.92  | 2.77  | 2.21  | 2.70  | 2.95  | 2.64  |
| Ours        | 2.07  | 1.89  | 1.91  | 1.92  | 2.27  | 1.94  | 1.94  | 1.90  | 3.54  | 2.02  | 2.14  | 1.89  | 2.62  | 2.16  | 2.64  |

[1] Qianli Ma, Jinlong Yang, Anurag Ranjan, Sergi Pujades, Gerard Pons-Moll, Siyu Tang, and Michael J Black. Learning to dress 3d people in generative clothing. In Proceedings of the IEEE/CVF Conference on Computer Vision and Pattern Recognition, pages 6469–6478, 2020.

[2] Matthew Loper, Naureen Mahmood, Javier Romero, Gerard Pons-Moll, and Michael J. Black. SMPL: A skinned multi-person linear model. ACM Trans. Graphics (Proc. SIGGRAPH Asia), 34(6):248:1–248:16, October 2015.

[3] Bharat Lal Bhatnagar, Garvita Tiwari, Christian Theobalt, and Gerard Pons-Moll. Multi-garment net: Learning to dress 3d people from images. In Proceedings of the IEEE/CVF International Conference on Computer Vision, pages 5420–5430, 2019.

---

### Decision · Program_Chairs · 2021-09-27

**Decision:**

Accept (Poster)

**Comment:**


This paper proposes a framework  to reconstruct 3D garment models from 3D point cloud sequences of dressed humans. The paper raised concerns regarding robustness of the model to segmentation errors, limited real world results, similarities to recent works, requirement for body point clouds (under the garment), and accuracy of the proposed method. The rebuttal submitted by the authors addressed these concerns by showing real world results and the feasibility of obtaining body pointclouds using the SMPL model. Though the relevance of the paper to a wide NeurIPS audience maybe still be under question,  all reviewers are positive about the paper, and it is suggested for publication.